Independent validation of downscaled climate estimates from a coastal Alaska watershed using local historical weather journals

Williamson Emily R. erwilliamson3@alaska.edu 1
Sergeant Christopher J. 1 2
1 College of Fisheries and Ocean Sciences, University of Alaska–Fairbanks , Fairbanks , AK , United States of America
2 Flathead Lake Biological Station, University of Montana , Polson , MT , United States of America
Raga Graciela
Electronic publication date: 2021 Sep 10
Publication date: 2021
Volume: 9
Electronic Location ID: e12055
Received 2021 Apr 1; Accepted 2021 Aug 4
Copyright: ©2021 Williamson and Sergeant
Copyright year: 2021
Copyright holder: Williamson and Sergeant
License: This is an open access article distributed under the terms of the Creative Commons Attribution License, which permits unrestricted use, distribution, reproduction and adaptation in any medium and for any purpose provided that it is properly attributed. For attribution, the original author(s), title, publication source (PeerJ) and either DOI or URL of the article must be cited.
License URL: https://creativecommons.org/licenses/by/4.0/

Keywords: Downscaled climate models, Alaska, Natural history, Precipitation, Air temperature, Hydrology, Climate change, Historical weather, ClimateNA, Validation

Funding: National Science Foundation award and by the State of Alaska OIA-1757348 This material is based upon work supported by the National Science Foundation under award #OIA-1757348 and by the State of Alaska. The funders had no role in study design, data collection and analysis, decision to publish, or preparation of the manuscript.

==============================
Downscaling coarse global and regional climate models allows researchers to access weather and climate data at finer temporal and spatial resolution, but there remains a need to compare these models with empirical data sources to assess model accuracy. Here, we validate a widely used software for generating North American downscaled climate data, ClimateNA, with a novel empirical data source, 20th century weather journals kept by Admiralty Island, Alaska homesteader, Allen Hasselborg. Using Hasselborg’s journals, we calculated monthly precipitation and monthly mean of the maximum daily air temperature across the years 1926 to 1954 and compared these to ClimateNA data generated from the Hasselborg homestead location and adjacent areas. To demonstrate the utility and potential implications of this validation for other disciplines such as hydrology, we used an established regression equation to generate time series of 95% low duration flow estimates for the month of August using mean annual precipitation from ClimateNA predictions and Hasselborg data. Across 279 months, we found strong correlation between modeled and observed measurements of monthly precipitation (ρ = 0.74) and monthly mean of the maximum daily air temperature (ρ = 0.98). Monthly precipitation residuals (calculated as ClimateNA data - Hasselborg data) generally demonstrated heteroscedasticity around zero, but a negative trend in residual values starting during the last decade of observations may have been due to a shift to the cold-phase Pacific Decadal Oscillation. Air temperature residuals demonstrated a consistent but small positive bias, with ClimateNA tending to overestimate air temperature relative to Hasselborg’s journals. The degree of correlation between weather patterns observed at the Hasselborg homestead site and ClimateNA data extracted from spatial grid cells across the region varied by wet and dry climate years. Monthly precipitation from both data sources tended to be more similar across a larger area during wet years (mean ρ across grid cells = 0.73) compared to dry years (mean ρ across grid cells = 0.65). The time series of annual 95% low duration flow estimates for the month of August generated using ClimateNA and Hasselborg data were moderately correlated (ρ = 0.55). Our analysis supports previous research in other regions which also found ClimateNA to be a robust source for past climate data estimates.

Introduction

Across many scientific disciplines, researchers rely on the downscaling of coarse global and regional climate models to access weather and climate data at finer temporal and spatial resolution (Mote & Salathé, 2010; Xu, Han & Yang, 2019). While weather stations collect accurate local measurements, they are not evenly distributed and leave many empirical data gaps across the globe. Statistical downscaling of weather measurements such as rainfall or air temperature, rather than empirical observations, is one of several methods available for analyzing climate and hydrologic response in remote regions where station maintenance is difficult and expensive. ClimateNA is a widely used, publicly available, and user-friendly software that produces a suite of statistically downscaled monthly and annual climate variables for point locations across North America that are dynamically adjusted for local elevation (Wang et al., 2016). The software’s simplicity and wide range of historical and future time ranges are well suited for correlating climate patterns with questions related to many environmental science disciplines. A recent study demonstrated that monthly temperature variables generated by ClimateNA performed well against measured data from 232 weather stations in southern Alberta, Canada, but some biases existed across space, season, and elevation (Roberts, Wood & Marshall, 2019). In our present study, climate data generated by ClimateNA provide the basis for our validation with a novel source of empirical weather data from sub-arctic Alaska. While global climate model performance for this region has been compared to re-analysis products that combine weather observations with numerical modeling (for example, see Herzfeld et al., 2007; Walsh et al., 2008), very few climate model validation efforts are based on independent empirical observations.

In Alaska, air temperatures are expected to rise over time alongside increases in annual precipitation and extreme precipitation (Lader et al., 2017). In remote regions such as our study location in Southeast Alaska, USA, downscaled projections of temperature, precipitation, and snowfall are critical to understanding regional hydrology and ecology (Bieniek et al., 2016; Littell, McAfee & Hayward, 2018). For example, combining river basin topography with estimates of monthly precipitation allows for the calculation of various hydrologic metrics such as percentile flow magnitudes (Wiley & Curran, 2003), which directly relate to the efficacy of burgeoning hydropower operations (Cherry et al., 2017) and the population dynamics of many aquatic organisms (Poff et al., 1997; Brown et al., 2016). The success of culturally and economically valuable fish populations in Alaska such as Pacific salmon (Oncorhynchus spp.) is inextricably tied to flow regimes (Schoen et al., 2017; Johnson et al., 2019); therefore, having access to high-quality climate information is critical for resource managers and users to assess habitat conditions over time and the potential for future change.

Climate change is occurring rapidly in Southeast Alaska. Regional experts predict increases in annual average air temperature and precipitation, with less precipitation falling as snow during the fall and winter (Shanley et al., 2015; Littell, McAfee & Hayward, 2018; Lader et al., 2020). A key challenge is determining the rate at which these changes are occurring, thus it is important to compare contemporary climate trends with historical data sets. Historical climate time series derived from direct measurement methods are often sparse in remote locations. In these instances, downscaled data can be used to estimate climate metrics across large scales of time and space, but their local accuracy is difficult to determine without validation using independent empirical data. This is especially challenging in topographically complex areas like Southeast Alaska, where mountains rise from sea level to hundreds of meters above sea level within a few km. This steep topography promotes orographic lifting, which raises moist air and condenses it into precipitation. A mosaic of mountains, glaciers, and narrow marine passages create dynamic micro-climates combining wet and mild coastal zones with drier and colder continental conditions (Shanley et al., 2015), making it difficult to discern the accuracy of downscaled climate patterns relative to nearby weather stations. For example, in a validation study of the North American arctic, modeled climate accuracy was variable across seasons and better at predicting air temperature than precipitation (Herzfeld et al., 2007). Since downscaling creates a high-resolution grid of interpolated data, the quality of digital elevation models used has a direct bearing on data quality. In steep terrain, the grid cell of a coarse elevation model may span hundreds of meters of elevation change and lead to misleading or difficult-to-interpret data outputs. ClimateNA addresses this issue through a combination of bilinear interpolation and elevation adjustments that allows users to extract data from a specific point in space that is not averaged over an entire grid cell (Wang et al., 2016).

Considering the difficulty in predicting climate patterns for a geographically complex region, the extent to which downscaled climate models can be corroborated with conventional weather measurements will improve the confidence of research and regional decision-making based mostly on modeling efforts. Here, we present a novel validation of downscaled climate data from 1926 to 1954 using archived weather journals from Admiralty Island, Alaska homesteader, Allen Hasselborg. Daily weather observations were collected by Hasselborg at his homestead adjacent to Mole Harbor and the Mole River, located on the eastern shoreline of Admiralty Island in Southeast Alaska (Figs. 1 and 2). During the years 1926–1954 covered by his journals, only 7 weather stations existed in the region, with the nearest in Juneau approximately 74 km away. Mole Harbor remains uninhabited to the present day.

Figure 1 Study area.

The Mole River watershed (gray polygon) and the location of Allen Hasselborg’s homestead (black dot).

Figure 2 The journals of Allen Hasselborg.

(A) Example pages from Allen Hasselborg’s weather journals. In the upper left corner of each page, note that Hasselborg summed total monthly precipitation in inches. In the third column of each page, Hasselborg provided a symbol describing weather conditions that day: clear (open circle), cloudy (cross), rain (upward facing arch), or snow (downward facing arch). (photo credit: lead author, ERW). (B) Hasselborg at his homestead next to Mole River in 1941 (Image accessed through the Alaska Digital Archives, https://vilda.alaska.edu/digital/collection/cdmg21/id/12774/rec/23).

Southeast Alaska is situated in the northern portion of the Pacific Coastal Temperate Rainforest, a generally mild and wet maritime landscape averaging 200 cm of precipitation per year (O’Neel et al., 2015). The maximum elevation within the 67 km2 Mole River watershed is 869 m while mean elevation is 303 m. With no glaciers present in its drainage area, the Mole River is a mostly forested watershed with discharge driven by a combination of rain and snow runoff (a “rain-snow-I” watershed as defined in Sergeant et al., 2020). Rain-snow-I watersheds are found throughout the coastal Gulf of Alaska. While maximum discharge tends to occur in mid-March for this watershed class, the flow regime of the Mole River remains unmeasured and maximum discharge could potentially occur during any day of the year (Sergeant et al., 2020).

Our primary objective was to determine how closely weather observations from Hasselborg’s journals correlated with monthly climate observations extracted from ClimateNA (Wang et al., 2016) at the same location and time periods. We then mapped the degree of correlation between the weather journals and modeled climate data across varying distances from the homestead site and elevations. We calculated residuals between observed and modeled values of monthly precipitation and monthly mean of the daily maximum air temperature to determine whether the accuracy of modeled data changed over time. To demonstrate the application of this research to other disciplines such as hydrology and fish biology, we compared streamflow metrics derived from the journals and modeled data.

Materials & Methods

Data acquisition

Data were acquired from an interactive climate model platform and Hasselborg’s handwritten weather journals. The widely used software, ClimateNA (Version 6.2; Wang et al., 2016), generates statistically downscaled climate estimates for any 4 × 4 km grid cell in North America, with customizable variables such as spatial coordinates, elevation, and date range. ClimateNA can generate precipitation and temperature data with monthly to annual resolution. The software incorporates and interpolates historical data spanning from 1901 to the present day, as well as a suite of general circulation models for future predictions. Hasselborg’s data were not used to generate historical data in ClimateNA and are therefore an independent source for model validation. We obtained monthly precipitation and monthly mean of the daily maximum air temperature values from the approximate GPS coordinates for Mole Harbor (57.647541, −134.094391). We allowed ClimateNA to determine elevation automatically. This provided model estimates overlapping with all of Hasselborg’s recorded years (1926–1954).

Hasselborg recorded daily temperature and precipitation between 1926 and 1954 at approximately sea-level near the mouth of the Mole River (Figs. 1 and 2). Missing time periods spanned July–September 1931, August–September 1932, June 1933, and December 1933 to January 1938. Hasselborg’s original handwritten weather journals are archived at the Alaska State Library and Archives in Juneau, Alaska, USA (Fig. 2). Journals were scanned as images and manually transcribed into a Microsoft Excel spreadsheet. The lead author (ERW) transcribed daily maximum air temperature (converted from °F to °C) and monthly precipitation (converted from in to mm). Hasselborg recorded precipitation in 0.25 in fractions, and recorded even and odd integers for temperature in his journals until 1933. It is possible that after a break from journaling from 1933 to 1938, Hasselborg used a second thermometer or changed his protocol for temperature measurements as nearly all post-1938 temperatures are even integers. To our knowledge, Hasselborg did not provide notes on his temperature collection methods. We also do not know if his thermometer was shielded from sunlight. We hypothesized that he collected maximum daily air temperature because this is the most logical consistent measurement that could be collected from an analog thermometer without having to collect measurements at the exact same time each day. Subsequent analyses in the Results section support this hypothesis. We excluded unreadable journal days or entries with multiple measurements. We also do not know for certain whether Hasselborg included snow in his precipitation amounts, but based on journal pages where he makes separate notes for rain and snow, we believe he intended to only measure rain. This does not exclude the possibility of some error due to snow falling into the measurement tool along with rain during winter months.

Correlation analysis

For monthly precipitation and monthly mean of the maximum daily air temperature measurements, we trimmed the ClimateNA data to account for Hasselborg’s data gaps (n = 279 usable months for precipitation, n = 274 usable months for temperature), and calculated a Spearman’s rank correlation coefficient (ρ) for the complete time series of empirical and modeled data. To assess spatial correlation at increasing distances from the Mole River watershed, we mapped the correlation between Hasselborg’s monthly precipitation data and ClimateNA datasets for the wettest year (1939) and the driest year (1951) found in his journals. Centroid coordinates for each grid cell in the spatial correlation map were spaced 0.2 decimal degrees longitude and 0.1 decimal degrees latitude apart and originated from the Hasselborg homestead.

To determine whether ClimateNA displayed seasonal or annual trends in estimating observed weather values, residuals were calculated across the entire time series by subtracting the monthly values for both precipitation and temperature measured by Hasselborg from those estimated by ClimateNA. To visually assess trends in the time series of residuals, we implemented a loess smoother, a non-parametric form of local regression where points are weighted based on their distance from the observation of interest (Jacoby, 2000). Residuals were also averaged across the four meteorological seasons: winter (DJF), spring (MAM), summer (JJA), and fall (SON).

Application of climate data sources to estimating streamflow descriptors

Low-duration streamflow was estimated using precipitation, watershed basin area, and watershed elevation in a regression equation derived from streamflow gage data (Wiley & Curran, 2003). To demonstrate the application of our validation to useful hydrologic estimates, we calculated the annual 95th percentile low-duration flow for August (AUG95) in ft3/s using the following equation: AUG95=1.397×10−9A1.16P1.367E1.896

Where,

A = watershed drainage area (mi2)

P = mean annual precipitation (in)

E = mean basin elevation (ft)

AUG95 estimates in ft3/s were then converted to m3/s.

The Wiley & Curran (2003) equation typically uses, “mean annual precipitation averaged over the drainage basin,” but in order to facilitate a more direct streamflow comparison between ClimateNA data and the Hasselborg journals, we only extracted mean annual precipitation each year from ClimateNA using the coordinates of the Hasselborg homestead. This approach assumes that modeled or observed precipitation amounts from the homestead site were comparable to average conditions across the basin. We did not estimate low-duration August flow during years where Hasselborg skipped >3 months of precipitation measurements. Hasselborg skipped one to three months per year for four of the included years in the data (1931, 1932, 1933, 1946).

Results

We found strong correlation (ρ) between the monthly precipitation time series generated from modeled (ClimateNA) and empirical (Hasselborg’s journals) climate data (ρ = 0.74, P << 0.001; Fig. 3). Residual values of modeled - observed monthly precipitation data ranged from 189 mm to −412 mm with a mean of −14.4 mm (Fig. 3). In general, residuals demonstrated heteroscedasticity around the zero line for the vertical axis, but there was evidence of a negative trend in residual values starting during the last decade of data (Fig. 3). Hasselborg’s precipitation values had a greater range than Climate NA, which consistently overestimated monthly precipitation when Hasselborg observed less than 50 mm and underestimated when observations were greater than 400 mm (Fig. 3).

Figure 3 Time series and residuals of modeled and observed precipitation data.

(A) Monthly precipitation estimated by ClimateNA (orange) compared with monthly precipitation measured by Hasselborg (blue). (B) Residuals of monthly precipitation. A loess smoothed trend line with span = 0.75 (blue) is included along with a dotted reference line at y = 0.

There was high correlation between monthly means of the maximum daily air temperature generated from ClimateNA and observed by Hasselborg (ρ = 0.98, P << 0.001; Fig. 4). The residual values of modeled - observed monthly mean of the maximum daily temperature data ranged from −3.06 °C to 10.44 °C with a mean of 0.77 °C (Fig. 4). Residuals demonstrated a consistent but small positive bias, with ClimateNA tending to overestimate air temperature relative to Hasselborg’s journals (Fig. 4).

Figure 4 Time series and residuals of modeled and observed temperature data.

(A) Monthly mean of the daily maximum temperature estimated by ClimateNA (orange) compared with Hasselborg (blue). (B) Monthly mean of the daily maximum temperature residuals. A loess-smoothed trend line with span = 0.75 (blue) is included along with a dotted reference line at y = 0.

When the distributions of precipitation residuals were summarized across meteorological seasons, ClimateNA was less accurate with a more precise range of values in the summer relative to the other seasons, when precipitation was generally underestimated and demonstrated greater variability (Fig. 5). The distributions of seasonal residuals of temperature demonstrated an opposite pattern to precipitation (Fig. 5). ClimateNA overestimated the most during winter—and to a lesser extent, spring and fall—while summer months were most closely correlated.

Figure 5 Boxplots of seasonal residuals.

Boxplots of residuals aggregated by meteorological season for (A) monthly precipitation and (B) monthly mean of the daily maximum temperature. Black lines in the middle of each box represent median values. The top and bottom lines of each box represent 75th and 25th percentiles, respectively. The ends of the whiskers represent 1.5 x the interquartile range. Individual letters along the horizontal axis represent months of the year.

The degree of correlation between weather patterns observed at the Hasselborg homestead site and ClimateNA data extracted from spatial grid cells across the region varied by wet and dry climate years (Figs. 6 and 7). Monthly precipitation from ClimateNA tended to be more similar to empirical data collected by Hasselborg across a larger area during wet years compared to dry years. During the wettest year recorded in Hasselborg’s journals, 1939, mean ρ = 0.73 (SD = 0.11) across all grid cells (Fig. 6). For the driest journal year, 1951, mean ρ = 0.65 (SD = 0.14) across all grid cells (Fig. 7). For both years, correlations between ClimateNA and Hasselborg’s data tended to be strongest in areas of low to mid-elevation. Correlations were weakest in high elevation areas and icefields across both years, but correlations were especially weak along the continental mainland during the dry year of 1951. During 1939, correlations were generally very high for grid cells south of the Hasselborg homestead.

Figure 6 Spatial correlation of precipitation during a wet year.

Correlation between monthly precipitation values generated by ClimateNA and Hasselborg weather journals during 1939, the wettest year recorded in the journals. Grid cell colors correspond to the magnitude of correlation defined in the legend. Grid cells in the spatial correlation map were spaced 0.2 decimal degrees longitude and 0.1 decimal degrees latitude apart and originated from the Hasselborg homestead.

Figure 7 Spatial correlation of precipitation during a dry year.

Correlation between monthly precipitation values generated by ClimateNA and Hasselborg weather journals during 1951, the driest year recorded in the journals. Grid cell colors correspond to the magnitude of correlation defined in the legend. Grid cells in the spatial correlation map were spaced 0.2 decimal degrees longitude and 0.1 decimal degrees latitude apart and originated from the Hasselborg homestead.

The time series of annual 95% low duration flow estimates for the month of August generated using ClimateNA and Hasselborg data were moderately correlated (ρ = 0.55, P = 0.007; Fig. 8). While the directional trends in both time series were comparable across time, similar to the spatial correlation maps (Figs. 6 and 7), ClimateNA estimates of low duration flow tended to be further from estimates using Hasselborg observations during dry years such as 1951.

Figure 8 Using modeled and observed data to estimate low streamflow descriptors.

Annual estimates of 95% low duration flow in August using observed Hasselborg data (blue) and modeled ClimateNA data (orange).

Discussion

With an ever-growing need to compare the changes in the earth’s climate from historical times to present-day, validation of climate models with empirical data will continue to be an essential aspect of environmental research. Historical weather measurements collected from remote locations have the potential to underestimate actual conditions (Baudouin, Herzog & Petrie, 2020) and be of questionable quality. To our knowledge, Hasselborg’s Mole River data set did not include a second observer or provide any methodological accounting. Despite these limitations, the corroboration of Hasselborg’s data and downscaled estimates produced from ClimateNA suggests that both sources are generally reliable. Since our analysis covers only a very small geographic area, we caution that our results should not be generalized beyond the northern portion of the Southeast Alaska panhandle. In western North America, there are many more examples of inter-model comparisons of climate data (for example, Jiang et al., 2018) instead of the validation with empirical data presented here. Although a validation of ClimateNA across broader spatial scales using additional empirical data was beyond the scope of our study, a logical next step would be to use data from other historical weather stations using similar methods. Reviewing publicly available records through the National Weather Service (https://w2.weather.gov/climate/xmacis.php?wfo=pajk), we determined the nearest weather station with temperature and precipitation data overlapping with Hasselborg’s journals was in Juneau, approximately 74 km away. Other overlapping data are available from Haines, Little Port Walter, Petersburg, Sitka, Skagway, and Yakutat and range 100-300 km from the Hasselborg homestead. We do not know the extent to which these data are independent from ClimateNA generated data.

Similar to the findings of Herzfeld et al. (2007) in the North American arctic, modeled air temperature data were more strongly correlated with observed data than precipitation. We also found support for the reliability of ClimateNA estimates in low-lying maritime environments throughout the northern portion of the Alaska Panhandle (Southeast Alaska), and correlations across space with Hasselborg’s data were especially strong during a wet year. In southern Alberta, Canada, Roberts, Wood & Marshall (2019) found that ClimateNA performed well against air temperature variables collected across 232 weather stations, but downscaled values tended to overestimate temperature at higher (>2000 m above sea level) and lower (<1000 m above sea level) elevations. A consistent theme in our work and others cited herein is that recognizing and potentially correcting model biases are critical for the practical application of climate downscaling in fields such as ecology (Roberts, Wood & Marshall, 2019). We are not aware of any other validations of ClimateNA using historical and independent empirical data.

The southern coastal region of Alaska is undergoing shifts in climate that impact the ecology of the region (Shanley et al., 2015). As a snow-and rain-driven system, the Mole River is representative of many low-lying watersheds throughout Southeast Alaska. Watersheds to the south and west of Mole River are primarily rain-driven, relatively lower elevation, and more exposed to maritime influence (Sergeant et al., 2020). This may explain the high correlation between Hasselborg and ClimateNA data in the southern portion of our study region during the wet year of 1939. Although adjacent watersheds should have similar climate patterns to the Mole River, we urge researchers to use caution when applying downscaled climate data to other ecological and physical estimates, especially in higher elevation watersheds with snow and glacier runoff patterns. While the August low streamflow descriptor we derived from empirical and modeled data followed the same general directional patterns over time, there were significant differences in the estimates of flow magnitude, especially during dry years such as 1951. This might be explained by long-range atmospheric drivers such as the Pacific Decadal Oscillation (PDO). The negative trend in modeled precipitation residuals during the final decade of data (1944-1954), correspond with a shift to a cold-phase PDO. This may be coincidental timing or it may be due to local variability in topographically complex terrain that is difficult to model correctly during oscillation shifts. In Southeast Alaska, the cold-phase PDO typically translates to more winter precipitation falling as snow than rain and resulting in higher-than-average summer discharge (Mantua et al., 1997; Neal, Walter & Coffeen, 2002). Patterns such as these are important context for researchers applying downscaled climate data and suggest that further validation research is necessary across a broader range of elevational and latitudinal gradients.

Supplemental Information

Supplemental Information 1 Monthly precipitation

Click here for additional data file.

Supplemental Information 2 Monthly air temperature

Click here for additional data file.

Supplemental Information 3 August low-flow

Click here for additional data file.

Supplemental Information 4 R code

Recreate Figs. 3, 4 and 5

Click here for additional data file.

We thank Uma Bhatt and Rick Lader for advice and helpful reviews of preliminary manuscript drafts. We thank David Roberts and Michael Winfree for thoughtful and constructive peer reviews that greatly improved the final manuscript. Sandra Johnston at the Alaska State Library and Archives provided access and scanning tools for Hasselborg’s journals.

Additional Information and Declarations

Competing Interests

Author Contributions

Data Availability

The authors declare there are no competing interests.

Emily R. Williamson and Christopher J. Sergeant conceived and designed the experiments, performed the experiments, analyzed the data, prepared figures and/or tables, authored or reviewed drafts of the paper, and approved the final draft.

The following information was supplied regarding data availability:

The raw measurements and the R code used to analyze these data are available in the Supplemental Files.

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
