# Peer review of "Independent validation of downscaled climate estimates from a coastal Alaska watershed using local historical weather journals"

_PeerJ, doi:10.7717/peerj.12055_

## Round 0.1 · original submission · Major Revisions

As you can see below, there is discrepancy in the recommendations made by the 2 reviewers: Accept versus Major Revision. My own assessment of your submission is that it requires the recommended revisions. So please carefully consider and respond to ALL the comments made by Reviewer 1.

·

Basic reporting

The writing is generally clear and concise. I do think that more literature on downscaling and validations of climate downscaling could be reviewed in the Introduction. It would establish more context for the novelty of this validation to know that most downscaling validations are either not done at all, or are done using non-independent cross-validation approaches.

While this study is a novel and useful evaluation of the ClimateNA downscaled outputs, I’m not sure that it’s a “cross-validation” per se. Whereas cross-validation is a resampling technique that entails data splitting and model training/evaluation with partial datasets, this study uses an independent dataset to validate or evaluate the ClimateNA data. My suggestion would be to change “cross-validation” to “validation” or “evaluation” throughout the paper. (To be honest, I think calling this a cross-validation actually sells the analysis short. Independent validation is, in many ways, the gold standard of model evaluation whereas cross-validation is what we settle for when we lack independent data.) The title is similarly a bit confusing in this respect. I would suggest something more direct (and compelling) like “Independent validation of downscaled climate estimates from a coastal Alaska watershed using local historical weather journals”.

It would be nice to see a bit more background on Allen Haselborg’s journals presented in the introduction. These are such a novel and rich historical dataset that it seems somewhat unjust to bury them in the Methods. I think the authors would be justified to move the entire “Study area description” from the Methods to the Introduction and simply start the Methods section with “Data acquisition”. Editorial advice might be useful here too.

On figures, I didn’t want to go deep down a rabbit hole of figure rebuilding, but I think there are simple changes that could easily be made to the figures to make them much more useful in terms of interpretation of the results. In fact, just my own plotting the data in different ways raised a few new questions/discussion points in my mind, and I think could form the basis for a bit more Discussion. I have provided some examples in the specific comments below.

Experimental design

The research question is well defined in the Introduction and is a useful contribution. As noted above, the novelty and importance of the research could be better highlighted with a more comprehensive literature review (which, as mentioned above, I’m confident would expose an important gap to fill that this paper helps address).

The streamflow calculation is an interesting addition, but it wasn’t totally clear to me how it was implemented across a larger region for the Hasselborg observations. I assume that the authors used the single data point from Hasselborg for temp & precip and the consequent differences in flow across the grid cells would be all due to elevation. In that case, were the ClimateNA streamflow calculations done based on a single data point at the Hasselborg cabin Lat/Long, or was data pulled from ClimateNA for each grid cell (in which case differences are due to differences not just in elevation but also in temperature and precipitation at each grid cell). This should be clarified. I think the two comparison options here ask fundamentally different questions.

Validity of the findings

All relevant data and R code were provided. I was able to run the code and confirm the results.

The Discussion is generally well written, but left me wanting more. It would be interesting and informative to have a bit more context for the results presented here. The most obvious context to me (given our 2019 paper) is a comparison to other validations of the ClimateNA software. I think the results presented here are largely in line with what we found in our analysis, which is exciting and would be interesting to discuss a bit further. I’m not suggesting this simply to highlight our own work (honest!). I’m just not aware of other similar validations of ClimateNA or other downscaling products, but that would be something also to discuss. Are other historical climate records available from the region? From elsewhere? Can we use this approach in larger-scale or wider-breadth validation efforts or is this really a one-off validation for this particular local area? The authors note that there were a few weather stations in the Mole River area (though not in close proximity to the Homestead, and I’m not certain on the periods of record). I would be interested to see if validations against these stations’ data showed similar patterns to the Hasselborg journals. Not suggesting this be added to the current manuscript, but could be put forward as a possible future analysis.

An obvious limitation of the current research is that it only considered a very small geographic region, so results should be generalised with caution (i.e. across the larger coverage of the ClimateNA software). This should be stated a bit more obviously I think. This point also offers an opportunity to discuss historic observation data as a potential source of validation across the continent for a more generalisable validation.

Additional comments

L40-43 – The abstract jumps a bit abruptly to reporting on stream flow. Suggest revising to include a general statement about calculating stream flow comparisons from the data, which may catch readers off-guard if they are anticipating a climatic comparison only.

L50-53 – A bit jargony. Would be useful to either (1) limit the discussion to statistical downscaling (what ClimateNA does) and explain what that is, or (2) provide a statement for each downscaling approach and identify which applies to ClimateNA.

L55-57 – So our 2019 paper doesn’t really support this statement (though I don’t disagree with it). More relevant to the paper here, results from our 2019 paper suggested that ClimateNA (in addition to being user-friendly and geographically flexible) produced downscaled data that were among the more accurate of those we considered (at least in our study area in the Rocky Mountains). BUT, like this paper, we also found biases in space and elevation. This paper looks more at biases in time, so the two analyses complement each other well. I also think our 2019 paper is useful to the current manuscript by providing some rationale for ONLY looking at ClimateNA in this validation, rather than at a suite of downscaling models.

L79-80 – Suggest: “but their LOCAL accuracy…”. Landscape/regional/continental scale validations (and cross-validations) of downscaled data are much more common using weather station networks (including during their development, as with ClimateNA).

L80-87 – This is an important section that would benefit from a bit more background on what statistical downscaling is and how it relates to topography (see comment on L50-53 above). Accurate DEM data are critical, even simply to implement ClimateNA, particularly at smaller scales/extents. The implications of topographically complex terrain will be more obvious if the reader understands the critical link between downscaling and DEMs.

L82 – Suggest “masl” or write out “metres above sea level” since we’re talking XY and Z distances in the same sentence.

L124 – Add the version of ClimateNA used (the algorithm and base datasets can change between versions).

L205-209 – This is a really interesting result. Potentially has to do with a bounding algorithm of some kind that may be built into ClimateNA (and other downscaling models) to prevent exponentially high/low predictions. Hopefully Tongli Wang is a reviewer on this paper also and can comment.

L215-216 – For context compared to another validation of ClimateNA, in Roberts et al. 2019 we found a similar trend in our comparison, with ClimateNA generally tending to overestimate monthly temperature, particularly at lower (<1000 masl) and higher (>2000 masl) elevations, and either highly accurate or slightly underestimating at middle elevations (1000-2000 masl).

L262-267 – The hypothesis about the PDO driving bias in ClimateNA is well defined here, but ClimateNA likely also incorporates such phenomena in its data/algorithm. It would be interesting to hear more about how this divergence might emerge, particularly at the very local scale at which the authors are working (again, hopefully Tongli is also a reviewer). In other words, perhaps ClimateNA does well with the PDO and other oscillations on a continental scale, but less well locally or during years of oscillation shift. I’m not sure, but a bit more explanation here of why the authors think PDO is the culprit (other than the coincidental timing) would be useful.

Figure 1 – This is a nice map of the river basin, but is a bit lost between the huge AK key map and scale/north arrow. Can those components be scaled down and the map scaled up? Hard to tell how big it will all be once sized for the journal, but I’m thinking one column wide? If that’s the case, the map will be almost too small to read.

Figure 4 – Could be nice to see the residual figure presented in the same orientation as Figure 3 (stacked, with date stretched across). I don’t get a lot out of the residual plots beyond the inflection of the loess trendline. I was trying to think if there’s a way to present the actuals and residuals together to offer a bit more interpretability. When I plot them together, the peaks & troughs in the residuals become much more evident in terms of timing and seasonality. Not necessary to use this approach, just a suggestion to make the residual patterns a bit more visually evident (also see next comment). Also suggest keeping at least the vertical panel axes so it’s easier to see seasonality within years. Same with temperature would also be useful.

[See alternative precipitation plot.]

An alternative to including the loess splines to demonstrate the deviation of residuals in later years might be to show something like the cumulative residual over time, which visualises bias changes in the model (ClimateNA) over time. This is not a *typical* way to present these data, but helps accentuate the divergence of residuals in later years without the need for a loess spline implementation. (On that note, the loess spline implementation needs a bit more background explanation/rationale if it’s going to be shown on the plot.)

[See cumulative residual plot.]

Figure 5 – I think using boxplots rather than plotting means and SD for these plots would increase the interpretability and identify other interesting patterns in the residuals. (Also, the precipitation residuals are somewhat negatively skewed, making means and SDs somewhat suboptimal.) For example, the tighter distribution of summer precip residuals is an interesting pattern worth exploring, and I would be interested in the severe outlier in spring temperature residuals (a point that is hidden by mean/SD plots).

[See boxplot.]

·

Basic reporting

Very clear and concise report. It was clear what your questions/hypotheses were and the results were relevant to the questions. Good use of references.

I have no suggestion for changes.

Experimental design

Excellent design. Good structure to lay out the results of your main objectives and a good example of applying in a hydrologic context.

Validity of the findings

No suggested changes.

Additional comments

I'm feel privileged to have the opportunity to review this paper as it was a fun to read. I don't have suggestions for major revisions. Of course, reading this led me to ask several questions about ClimateNA - which is to say your point about needing "further cross-validation research" is highlighted and this research provides a good model to build future research on.

---

## Round 0.2 · accepted · Accept

Your revised version addressed the comments made by reviewers and I consider that the final manuscript better highlights the study and reads very well.

·

Basic reporting

Very good. No further comments.

Experimental design

No comments.

Validity of the findings

No comments.

Additional comments

I think this version of the mansucript is much improved. While the science was robust in the previous version also, the additions to the Intro and Discussion specifically help highlight the novelty and importance of the work, and the clarity added around the methodology is welcome.